# COVID-19 and Behavioral Factors of e-Payment Use: Evidence from Serbia

Miloš Milosavljević [1,*], Milan Okanović [1], Slavica Cicvarić Kostić [1], Marija Jovanović [2] and Milenko Radonić [1]

1  Faculty of Organizational Sciences, University of Belgrade, 11000 Belgrade, Serbia
2  Faculty of Pharmacy, University of Belgrade, 11000 Belgrade, Serbia
*  Correspondence: milos.milosavljevic@fon.bg.ac.rs

**Abstract:** Banknotes and coins are some of the most frequently traded items in the world. Their current use, however, is unsustainable, and many countries are trying to digitalize their payment systems. The recent pandemic has accelerated this transition. Building on the Theory of Unintended Consequences, the aim of this article is to examine the influence of some pandemic-specific factors (in specific, hand sanitization, conspiracy theory mentality, and financial acumen) on the current and prospective use of e-payment. A particular aim of the study is to analyze these relationships in Serbia (as an example of a cash-centric society). The study is based on primary data gathered via a questionnaire. The questionnaire was designed for the purpose of this study. In total, the study examined N = 474 examinees. The results of this study confirm that the pandemic-induced variables are statistically significant predictors of e-payment use. In particular, hand sanitization, conspiracy mentality (reversely), and financial acumen positively affect current and prospective e-payment use.

**Keywords:** COVID-19; sustainable finance; e-payment; pandemic; technology acceptance

## 1. Introduction

Banknotes and coins are some of the most frequently traded items in the world. This was an introduction to a study conducted by Gedik, Voss, and Voss [1], who examined the transmission of disease-causing bacteria on banknotes from different countries. This study might not come as a surprise now, but at the time this finding was published (the year 2013), the authors became the laureates for the mockingly named scholarly award—the Ig Nobel Prize. The context, however, has dramatically changed ever since. The coronavirus pandemic has become an unprecedented health concern globally. As of November 2022, the number of confirmed cases had passed 639 million internationally, and sadly, more than 6.6 million people had died around the globe from this severe disease. An insight into the transmission of disease-causing microorganisms posed by the 'laughed-at' authors might come back as a boomerang nowadays.

Since the earliest days of the outbreak of SARS-CoV-2, virologists, epidemiologists, and other medical professionals have been trying to identify the causes, nature, and means of virus transmission. Naturally, the respiratory droplet or airborne route of any infectious disease has the highest potential to disrupt social interactions [2], let alone traditional payments. The WHO and governments worldwide have been trying to persuade the global and local populations that the use of banknotes and coins is safe. However, some sanitation measures to sterilize cash were used as a preventive measure, particularly in China (i.e., [3]). It seems that banknotes and coins will be less appealing to their frequent users due to the increased health concerns. Although this interrogative is close to common sense, only a paucity of research radars has hitherto been directed at answering it [4,5].

On the other hand, the ongoing pandemic, combined with decentralized-democratized social media proliferation, has brought forth yet another pandemic peculiarity—an increased belief in fake news and conspiracy theories [6–8]. We do not argue that the pandemic gave birth to a myriad of conspiracy theories (i.e., the Flat Earth Society has

been present globally for many years now). We additionally do not argue that media coupling with governments worldwide has not made negative information and nocebo effect, as already pointed out in recent research [9]. However, the pandemic has most certainly fostered and accelerated the diffusion and differentiation of fake 'news' and theories. However, it has most certainly fostered and accelerated the diffusion and differentiation of such 'news' and theories. The motivation behind this is to build a case for the survival of paper and metal cash. The line of reasoning is straightforward—cash transactions cannot easily be traced [10]. With fake news and conspiracy theories circulating in the media, trust in governments, authoritative institutions, and organizations is declining heavily. Accordingly, we argue that in such circumstances, people might change their cash usage regime. Since it would be impossible to measure the belief in conspiracy theories (at least the authors of this study are not entitled to isolate what is and what is not a conspiracy theory), we used a proxy indication—the conspiracy mentality.

The pandemic has had another important impact—financial constraints for many people. As inferred from previous studies, people seldom make unwise financial decisions [11]. People also individually differ in terms of the extent to which they worry about financial issues [12]. During the pandemic, financial acumen increased as an expected outcome of a potential or real financial and economic crisis [13].

Even though all the pandemic-induced factors might propel or discourage the use of cash (or alternatively, encourage or constrain the use of e-payment), these factors have not been examined in the concurrent body of knowledge. Although this paper generally seeks to find the causality between the pandemic-induced factors and e-payment use, we take a rather neutral stance when it comes to the overall economic, political, or societal effects of the proliferation of novel technologies in payment services. We recognize both the positive sides and potential pitfalls of payment technology [14].

In the grand scheme of things, the drivers of e-payment use certainly not a novel topic. Several studies have addressed the issues of individual factors influencing the use of electronic payments or other digital financial services [15,16]. However, the way in which a pandemic affects e-payment use might attract both scholarly and general attention. The aim of our study is to analyze the influence of the pandemic on the future use of cash and inspect possible routes for the development of e-payment and cashless societies. We aim to explore how lockdown/pandemic-driven factors such as health concerns, the rise of conspiracy theories, and the outcomes of investments in financial knowledge, skills, and attitudes might affect e-payments in the long run. For this purpose, we collected primary data using a questionnaire as a research tool. We collected the responses in Serbia, which is still considered to be a cash-centered country [17,18].

To the best of our knowledge, a study of this kind has never been conducted before. From a broad perspective, several studies have examined the influence of pandemics on e-payments [19,20]. However, the factors described in this study were largely neglected. As for health concerns, a paucity of studies has tangentially examined the possible influence of the COVID-19 pandemic on the use of 'dirty money' [4]. As for the second objective of this study, not even a timid and circumlocutory scholarly attempt has been made to put the spotlight on cash use over the conspiracy theory/fake news nexus. As for the third objective, even prior to the pandemic, financial acumen and intelligence were examined as focal factors of e-payment use. It has been latently observed as a technology acceptance driver [21] or a means of satisfaction for e-payment [22].

The remainder of this paper is organized in the following order. Section 2 starts with a literature review and puts a special emphasis on e-payment use and pandemic-specific factors with a high potential to disrupt traditional cash use. Section 3 delineates the methodology used in this study by explaining the hypotheses, research instrument, measures, variables, and sampling procedure. Section 4 outlines the results of our study. Section 5 contextualizes the results by comparing them to the findings of other authors and explains the key findings, implications, and limitations, together with some recommendations for further research.

## 2. Theoretical Background and Literature Review

In this section, we explain the underlying theory (the Theory of Unintended Consequences) and review the literature related to the relationship between the observed hypothesized pandemic and specific factors as potential determinants of e-payment use.

### 2.1. Theoretical Background

E-payment is certainly not a novel topic in scholarly literature. Most of the growing body of knowledge has been based on various technology acceptance theories, such as the Unified Theory of Acceptance and Use of Technology, the Technology Acceptance Model, and the Theory of Planned Behavior. These theories, nonetheless, only allow for 'low-tension' research as they advocate for the modelling of common-sense factors of novel technology use [23]. As the concept of this study is to capture pandemic-induced factors, we built our study on the Theory of Unintended Consequences.

In particular, this socio-technological theory posits that individual actions cause the recurrence of unacknowledged structures. Individuals self-consciously reproduce routine practices, which in turn produce unconscious effects on technology use [24]. Accordingly, different routines can alter technology use [25].

In this section, we first delineate the use of e-payment technologies and later build hypothesized relations between pandemic-induced factors and the use of novel payment technologies.

### 2.2. Literature Review

With the exponential growth of e-commerce, traditional payments using cash or cheques are becoming obsolete. Most of the functionalities of traditional cash have been replaced with e-payments, such as wire transfers, debit and credit cards, electronic fund transfers, and mobile payments [26]. E-payment can be defined as any use of electronic means by a payment processor, including the internet and mobile networks, for financial transactions between two entities.

E-payment has been rapidly developing around the globe. For instance, WorldPay [27] reports that digital/mobile wallets will become the most widely used method of e-payment with 41.8% and with an estimated growth of 11.4% by 2023. At the moment, regular point-of-sale (POS) payments have seen cash as the most dominant payment method, accounting for nearly a third of all transactions, followed by debit cards, digital wallets, and credit cards.

As for the business case of Serbia, the country has recorded constant growth in the volume of online payment transactions in the period 2013–2019 with a Compound Annual Growth Rate (CAGR) of over 30%. The total annual value of transactions processed online has also shown a positive trend, with (CAGR of 25%) during the same period [28]. Nevertheless, Serbia is far from being a highly digitalized country, as most transactions must be administered in the 'old-fashioned', analogue manner, including payments [29].

However, the use of e-payment might be accelerated with the pandemic. Pandemics have altered the way in which people make social relations such as payments, but we will be focused on three factors that theoretically have the largest impact on payment habits:

(1) The transmission of pathogenic microorganisms via banknotes and coins. Several studies worldwide imply that banknotes and coins have a potential role in the transmission of microorganisms [1,30]. Many of these studies were carried out in a laboratory under experimental conditions [1]. However, the type of isolates on the money in circulation, in addition to the method used, may be affected by various factors, such as the season, type of money, environmental conditions, or local community flora [31]. In the review paper of Angelakis et al. [20], the authors summarized the findings of 'dirty money' studies and concluded that bacteria such as 'Staphylococcus aureus, Salmonella species, and Escherichia coli can commonly be found on banknotes, whereas Escherichia coli, Salmonella species, and viruses, including the human influenza virus, Norovirus, Rhinovirus, Hepatitis A virus, and Rotavirus, can be transmitted through hand contact.' Hence, contaminated banknotes

or coins may pose a public health risk, especially when associated with the simultaneous handling of food, and may contribute to the spread of nosocomial infections [31].

Alongside the recognition of the potential transfer of pathogens, scientists have been proposing different means and measures to mitigate public health risks. An interesting strategy, for instance, is the use of nanofiber coatings for banknotes, which seems quite futuristic now [32]. Nonetheless, the most recommended strategy is hand hygiene [33]. It would be impossible for someone to sanitize hands with every contact with cash. Accordingly, contactless payment and other means of e-payment have been publicly advocated in high-risk environments [34].

(2) Conspiracy theories and the use of e-payment. Equidistant to the worldwide spread of COVID-19, the world has experienced an 'infodemic' in which conspiracy theories and fake news are being shared at a disquieting rate [35]. According to the network of researchers COMPACT, which comprises 150 academics across Europe who investigate the causes and consequences of conspiracy theories, these theories, which include the belief that behind-the-scenes events are run by powerful forces, are based on the assumption that nothing happens by chance, that things are not as they seem, and that all events are interconnected [36]. The importance and popularity of these theories have been growing steadily, with the potential to jeopardize and erode trust in medical, scientific, and even global authorities.

The study of Oleksy et al. [37] infers that people tend to believe in conspiracy theories in times of crisis. The COVID-19 crisis has generated a myriad of conspiracy theories that have been circulating worldwide and providing some sort of explanation for phenomena that deeply impact people's lives. Popular theories vary from those involving Bill Gates as one of the responsible ones for the creation of the virus in order to capitalize on a vaccine, over those theories connecting Coronavirus with 5G networks, explaining that the use of the virus with electromagnetic radiation aims at the intentional selection of the world's population, to those theories explaining the pandemic as an accidental spill from the Wuhan laboratory during the research on a biological weapon [38].

The impact that conspiracy theories might have on people can be elucidated by using the theoretical framework of reasoned action set by Fishbein and Ajzen [39]. The theory explains that the beliefs and attitudes of individuals (that can be formed based on experience and observation, or on information from external sources as well as on the reasoning process) drive their intentions and behaviors of these individuals. Accordingly, as conspiracy theories are external sources that provide information on a particular subject, they can lead to a particular behavioral change as well.

A relationship between the predisposition of people to form irrational beliefs (a set of beliefs, such as conspiracy or anti-science beliefs and attitudes, lacking a solid evidence base) and their tendency to go along with proper preventive measures for COVID-19 is examined in Teovanović et al. [40]. The authors find that belief in conspiracy theories related to COVID-19 was the most consistent predictor of health behavior, which has also been grounded in previous research that reports its positive relations with the use of pseudoscientific practices (such as consuming large quantities of garlic), not cohering with medical or public health recommendations, and reluctance to get vaccinated [40]. This leads to the conclusion that conspiracy theories can have unfavorable social consequences during the COVID-19 pandemic.

Aside from other social consequences, we posit that fake news and conspiracy theories might impact trust in the whole financial system, particularly in the realm of e-payments. Concurrent studies have been centered around payer-processor (bank) trust issues [41]. However, some studies extend this relationship to the financial system as a whole and confirm a positive relationship between trust in governments and the adoption of e-payment [42].

(3) Financial acumen and the use of e-payment. Both health concerns and the adoption of fake news would be irrelevant with the appropriate financial skills, knowledge, and

attitudes—put simply, financial acumen. Financial acumen is an amorphous term. It is sometimes referred to as financial intelligence [43] or financial literacy [44].

Terminology aside, financial acumen is generally propounded as an important predictor of an individual's wellbeing [45]. Recent studies even specifically confirm a positive relationship between the financial acumen of individuals and their affection for e-payments [46]. Given that financial acumen naturally grows in times of economic crises and constraints [47], it would hardly be judicious to assume that the COVID-19 pandemic (and the decline in economic activities worldwide [48]) will increase financial acumen. This, in turn, could positively affect the use of e-payments.

## 3. Materials and Methods

In this section, we explain the study hypotheses and the research instrument used to collect the data.

### 3.1. Hypotheses

To address the aims and objectives of the study and following the literature review, we tested the following hypotheses, as illustrated in Figure 1.

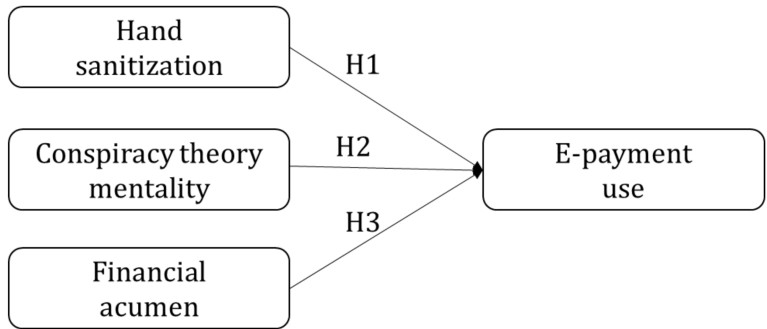

**Figure 1.** Hypothesized model for the pandemic-specific factors of e-payment use.

### 3.2. Research Instrument, Variables and Measures

Our study was based on primary data collected via an electronic questionnaire. The technique used in the study was CAWI (Computer-Assisted Web Interviewing). We acknowledge potential biases in the use of electronic questionnaires, particularly when asking sensitive or delicate questions related to personal hygiene or attitudes toward conspiracy theories. The more tactful approach to our respondents would be a hard copy (Paper-And-Pencil Inter-viewing) approach. Nonetheless, the main rationale behind the use of an electronic rather than a hard copy questionnaire was not time or cost efficiency [49], but rather momentum. The data was collected in the aftermath of the third COVID-19 wave, and this was the safest means of data collection.

The questionnaire had four distinct parts. The first one was aimed at collecting data on the demographic features of the respondents.

The second one was aimed at collecting evidence on hand sanitation after the use of cash. For this purpose, we selected self-reported practice. Using a 'self-reported hygiene practice' is common in novel prevention medicine and public health studies. For this particular purpose, specific scores such as the 'Handwashing Habits Score' were developed [50] long before the pandemic. However, scores such as this one have attracted immense attention after the breakout of COVID-19 (i.e., [51]). Since our study was not aimed at addressing overall hand hygiene or the efficient practice of it but only the need to sanitize hands after handling cash, we narrowed the measure to a singular dimension—hand hygiene when in contact with cash. The items used in the study are as follows: (1) I find cash to be unhygienic or dirty (HS_1); (2) How often do you wash hands after handling cash, banknotes, or coins (HS_2); and (3) How willing would you be to make a small change in your lifestyle to be more hygienic (HS_3). All the items were measured using a Likert-type scale.

The third part of the questionnaire was intended to measure the conspiracy mentality. Although every popular conspiracy theory attracts different individuals, many scholars have advocated that there is a stable spill-over effect. For instance, Swami, Chamorro-Premuzic, and Furnham [52] infer that if a person believes in one conspiracy theory, there is a great chance that he or she will believe in other conspiracy theories as well. Consequently, Bruder et al. [53] developed a Conspiracy Mentality Questionnaire, 'an instrument designed to efficiently assess differences in the generic tendency to engage in conspiracist ideation within and across cultures.' This scale has been rarely used ever since in various psychological studies [54–56]. For the purpose of this paper, we have used the translation of this questionnaire into Serbian deposited into the REPOPSI repository based on the study of Lukić, Žeželj, and Stanković [57]. The original scale includes the following items: (1) many very important things happen in the world that the public is never informed about (CMT_1); (2) politicians usually do not tell us the true motives for their decisions (CMT_2); (3) government agencies closely monitor all citizens (CMT_1); (4) events that superficially seem to lack a connection are often the result of secret activities (CMT_4); and (5) there are secret organizations that greatly influence political decisions (CMT_5). All the items were measured using a Likert-type scale. The items were reverse-scaled within data processing.

The fourth part of the questionnaire was aimed at measuring the financial acumen of respondents. Although there are a myriad of different approaches to the measurement of financial knowledge, skills, and attitudes, we awarded some inquiries from ref. [58] and developed a couple of our own. The items of financial acumen measurement are shown in Table 1.

**Table 1.** Financial acumen questionnaire items [the correct answers are in bold].

| Area | Question | Given Answers |
|------|----------|---------------|
| Investment | If you invest $100 in a savings account at 10% p.a., how much will you have at the end of year 2? | a. $100 ǀ b. $120 **c. $121** ǀ d. $128 |
| Transactions | Payment by credit card does not incur any costs | True-**False** |
| Banking | Investment in a single company's stock provides a safer return than an equity mutual fund | True-**False** |
| Taxation | John has $100 of gross honoraria. If the tax base is 50% and the tax rate is 10%, how much will John get paid net? | a. $50 ǀ b. $65 c. $90 ǀ **d. $95** |

The answers were coded in the following manner: the response was coded with 5 if all four questions were answered correctly; the response was coded with 4, if three out of four questions were answered correctly; . . . ; the response was coded with 1 if none of the answers were correct.

The fifth part of the questionnaire was aimed at measuring the dependent variable—the adoption and use of e-payments. Different scales are used to measure this phenomenon. Based on the Technology Acceptance Models, Riskinanto, Kelana, and Hilmawan [59] adopted a five-item scale. However, the best fit for our study is the scale developed by Kim et al. [60]. The items for the dependent variable were: (1) I use the e-payment system (EPS) more often than others (Epay_1); (2) I am currently using and will continue to use EPS (Epay_2); and (3) I believe EPS use will increase during and after the pandemic (Epay_3). All the items were measured using a Likert-type scale.

Sampling procedure and sample features: as the aim of the study was to examine the knowledge, attitudes, and behaviors of the general population related to the use of e-payment as a technology, we used a convenience sampling procedure. We tried to include respondents from various demographic groups, and accordingly, the questionnaire was iteratively delivered and coded to decrease possible invasive sub-clustering (see [61]). In total, the questionnaire was sent to 1.214 email addresses. We collected 486 responses, whereas 14 were marked as invalid. The criteria for exclusion were: a) less than 80% of

the questionnaire was filled; or b) the respondent did not answer the fourth part of the questionnaire (financial acumen questions). Accordingly, 474 responses were marked as valid, making the response rate 38.95%. The response rate is relatively high, but in line with other studies based on convenience sampling procedures [62].

## 4. Results

In this section, we elaborate on the study results. First, we explain the sample features and pre-analysis (including frequencies and descriptive statistics). Furthermore, we test the study hypotheses (using correlation and regression analyses).

### 4.1. Sample Features

In total, we have collected 474 responses. The average age of the respondents was 32.9 years, and the sample was slightly imbalanced in terms of gender, as 60% of the respondents were female. Half of the respondents in the sample were highly educated people, and 20% were currently enrolled in studies. The largest number of the respondents marked themselves as financially independent persons (65.4%), and most of the respondents (79.8%) assessed their technological competencies as good or great.

### 4.2. Pre-Analysis

Prior to testing the study hypotheses, we conducted a pre-analysis. Accordingly, we conducted descriptive statistics, internal reliability for multi-itemed constructs, and a correlation analysis for mutual interdependence between the observed variables.

As for the dependent variable, by analyzing the items related to e-payment, the respondents were the least in agreement regarding the use of e-payment (Mean = 3.72; SD = 1.38), which is somewhat expected, having in mind the demographic structure of the sample. However, the respondents were more coherent when it comes to the use of e-payment in the future (Mean = 4.00; SD = 1.10). The descriptive statistics for both independent and dependent variables are shown in Table 2.

**Table 2.** Descriptive statistics for multi-itemed measures.

| Item/Measure | Mean | SD | Item/Measure | Mean | SD |
|:---:|:---:|:---:|:---:|:---:|:---:|
| Epay_1 | 3.72 | 1.38 | CMT_1 * | 4.27 | 0.98 |
| Epay_2 | 3.93 | 1.24 | CMT_2 * | 4.51 | 0.93 |
| Epay_3 | 4.00 | 1.10 | CMT_3 * | 3.56 | 1.14 |
| E-payment use | 3.88 | 1.07 | CMT_4 * | 3.81 | 1.03 |
| HS_1 | 3.67 | 1.17 | CMT_5 * | 3.74 | 1.14 |
| HS_2 | 3.70 | 1.07 | Conspiracy Mentality * | 3.98 | 0.81 |
| HS_3 | 4.11 | 0.88 | | | |
| Hand sanitization | 3.83 | 0.74 | | | |

\* Reversely scaled.

The only independent variable not measured with the Likert-type scale was Financial Acumen. Instead, it was a sum of points given for the correct answers to four financial inquiries. Figure 2 displays the result of all the inquiries. The respondents were highly aware of financial transactions (58.44% of respondents answered correctly), whereas taxation was the main issue (only a quarter of respondents answered correctly).

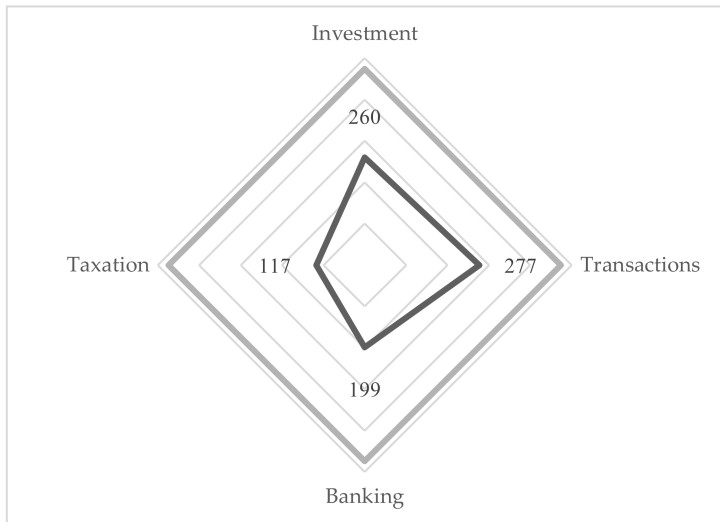

**Figure 2.** Four dimensions of the financial knowledge of respondents (investments, transactions, banking & taxation).

*4.3. Main Analysis*

Prior to testing the hypotheses, we examined the internal reliability of the multi-item constructs. The result is shown in Table 3. The reliability test for the scales used to examine the level of e-payment and Conspiracy Mentality showed high values for Cronbach's Alpha, 0.82 and 0.83. For the standard threshold of 0.70, we can firmly state that these multi-item measures show high internal reliability, which was expected as these measures are either rewarded or derived from the existing scales. Surprisingly, the value for Cronbach's Alpha for the variable Hand Sanitization fell beyond this threshold. Accordingly, we conducted an additional test—the inter-item correlation [50], and the value was an optimal value of 0.28. This provided the conditions for further work with composite variables for all three observed phenomena.

**Table 3.** Pre-analysis: means, standard deviations, internal reliabilities and correlation matrices.

| | | **Mean** | **SD** | **CA** | **2** | **3** | **4** |
|---|---|---|---|---|---|---|---|
| 1 | Hand sanitization | 3.83 | 0.74 | 0.28 [1] | 0.16 ** | −0.09 * | 0.13 ** |
| 2 | Conspiracy Mentality | 3.98 | 0.81 | 0.83 | | 0.02 | 0.25 ** |
| 3 | Financial Acumen | 2.80 | 1.17 | n/a | | | 0.22 ** |
| 4 | E-payment use | 3.88 | 1.07 | 0.82 | | | |

[1]—inter-item reliability (instead of Cronbach's Alpha); * $p < 0.05$; ** $p < 0.01$.

To identify the interdependence of the observed phenomena, we conducted a correlation analysis (Pearson correlation), and several statistically significant correlations were found (Table 3). E-payment positively correlates with all three independently observed variables: Hand Sanitization ($r1 = 0.13$; $p < 0.00$), Conspiracy Mentality ($r2 = 0.25$; $p < 0.00$), and Financial Acumen ($r3 = 0.22$; $p < 0.00$).

As we found a significant correlation between independent variables, we conducted multicollinearity diagnostics. The value for Durbin-Watson was 1.87, and the Variance Inflation Factor had low values (between 1.01 and 1.04). Accordingly, we concluded that no multicollinearity was found.

Afterward, we tested the study hypotheses. For this purpose, we used OLS regression. The influence of Hand Sanitization, Conspiracy Mentality, and Financial Acumen on e-payment was analyzed using the multi-variate regression model (Table 4).

**Table 4.** Regression model for E-paymelent Use as a dependent variable.

| | Unst.Coeff | | St.Coeff. | t | Sig. | VIF |
|---|---|---|---|---|---|---|
| | B | SE | Beta | | | |
| (Constant) | 1.45 | 0.33 | | 4.35 | 0.00 | |
| Hand Sanitization | 0.17 | 0.06 | 0.12 | 2.69 | 0.01 | 1.04 |
| Conspiracy Mentality | 0.30 | 0.06 | 0.23 | 5.20 | 0.00 | 1.03 |
| Financial Acumen | 0.20 | 0.04 | 0.22 | 5.13 | 0.00 | 1.01 |
| | R | 0.35 | Adj $R^2$ | 0.12 | DW | 1.87 |
| | $R^2$ | 0.12 | SE | 1.01 | F | 21.81 |

The OLS regression model (F = 21.81, *p* < 0.00) explained 12% of the variability of the dependent variable (R = 0.35, $R^2$ = 0.12), and all three independent variables participate in the model ($\beta_{HS}$ = 0.12; $\beta_{CM}$ = 0.23; and $\beta_{FA}$ = 0.22, respectively). Accordingly, we confirmed that all three independent variables (hand sanitation, conspiracy mentality, and financial acumen) have a statistically significant influence on e-payment use. This further implies that one-percent movement in independent variables (hand sanitization, conspiracy mentality, and financial acumen) causes 0.12% of the movement in the dependent variable (e-payment).

## 5. Discussion and Conclusions

### 5.1. Key Findings

The aim of our study was to analyze the influence of the pandemic on the use of e-payment in Serbia. Building on the Theory of Unintended Consequences, we tested three hypotheses. By examining 474 respondents, we confirmed the positive effects of the pandemic-induced factors on e-payment use. Although we only confirmed 12% of the total variability, these findings support the general position that the pandemic (in line with its consequences such as travel bans and lockdowns) might propel the use of digital payment services.

First, we confirm that hand sanitization might have a long-term impact on the abandonment of traditional cash. Several studies have confirmed that banknotes and coins may have a role in the transmission of multi-drug-resistant pathogenic microorganisms [1,31]. Similar to the other studies, we confirm that general awareness of paper and metal money as a fomite might cause various social effects [63], in our case faster adoption of e-payments.

Second, we confirm that fake news and conspiracy theories affect e-payment use. In other words, fake news and conspiracy theories deteriorate trust in the system, which affects the readiness to use e-payment. Other studies also confirm the positive relationship between individuals' trust and their readiness to use e-payment [64]. We extend this relationship by using a general conspiracy theory mentality as a proxy for trust in the system as a whole, thus providing a socio-economic rather than technical characteristic of trust.

Finally, we confirm that financial knowledge plays a statistically significant role in the use of e-payment. This has been confirmed by similar research conducted by Andreou and Anyfantaki [65]. It would be expected for financial acumen and knowledge to grow over time alongside the economic meltdown induced by the COVID-19 pandemic.

### 5.2. Contributions

This study contributes to the existing body of knowledge in both theoretical and practical areas. As for the theory, the prime novelty of this study is the redesign of the technology acceptance paradigm. Most of the concurrent studies have examined 'intended' antecedents of technology use and factors such as efficiency, effectiveness, effort expectancy, or social influence [66]. However, people tend to use digital services differently in times of crisis. Accordingly, our study reflects on the unexpected or unintended outcomes of the COVID-19 pandemic on technology use, thus extending the Theory of Unintended Consequences. Other studies also confirm that the COVID-19 pandemic has altered payments and

infer that we might see business bring us to "a fully digital lifestyle and potentially bringing us toward a potential cashless economy" [67]. Our study fills the lacuna in the concurrent body of knowledge by analyzing three new pandemic-induced factors of e-payment use. We examined how hand sanitation in contact with traditional cash (banknotes and coins), the explosion of fake news as a proxy for trust in e-payment system enablers, and a growing financial acumen affect the intention and future considerations related to e-payment use. These factors were outside the scope of other similar studies.

Second, our study examines specific factors in e-payments, an area that has developed immensely in the last few decades. It might seem speculative to claim that these payments have enabled the fulfillment of several personal needs and the survival of businesses around the globe during the time of COVID-19. Thereafter, this area is still a promising field of research [68], and the findings of our study would greatly contribute to the evolving body of knowledge.

Third, the study provides a theoretical contribution in geographic terms. Hitherto, only scarce evidence on e-payments has been collected in Serbia [69]. It has been more than two decades since the first bank in Serbia offered e-payment services. Since then, the banking system in Serbia has seen a great opportunity in digitalization and investing in digitalization, including e-payment systems. However, scholarly findings have not kept pace with the development of e-payment practices. Our study reflects on people's readiness to use e-payments and provides evidence of the pandemic-induced factors with the potential to accelerate their usage.

### 5.3. Implications

As for the practical values, this study can be of interest to several stakeholders, particularly policyholders and decision-makers from the financial ecosystem. Having in mind that the Serbian payment system is fully harmonized with EU legislation [70], it would not be a judicious judgement to claim that the findings of our study might be a fruitful source of information for both regulators and payment system operators beyond the borders of Serbia. More specifically, the stakeholders can leverage the COVID-19 pandemic to move towards cashless societies. This would certainly decrease the time and costs related to issuing, handling, and storing banknotes and coins. Although no country has yet become fully cashless, this move might decrease financial crime rates and prevent money laundering worldwide.

Again, this study takes a neutral position when it came to advocating e-payment use. The main beneficiary of the digital transition to electronic means of payment is the regulator in each country. Thus, this study might be interesting for financial system regulators to capitalize on the behavioral changes induced by the pandemics.

This study might be beneficial for businesses as well. Further proliferation of e-payment is changing customer behavior. Other studies also recognize that the use of cash has become a more costly solution [71] and that e-payment increases the financial awareness of customers [72].

### 5.4. Limitations and Further Recommendations

Although this study brings novelties to the present body of knowledge, its limitations simultaneously expose a number of opportunities for further research. First, we only examined a scarce number of pandemic-specific factors that affect e-payment use. It would be instructive for new studies to include other pandemic volatile behavioral personal characteristics that might exert influence on the proliferation of novel types of financial transactions. For instance, the locus of control, the effects of lockdowns, and travel bans might be interesting research opportunities. Second, this study is geographically constrained, and the results come from a single country. The findings might easily be generalized to other European countries with similar economic structures, history, and cultural backgrounds (i.e., other South-East European countries). Extending the findings beyond this geographical context would be nothing more than speculation. This provides

an opportunity for follow-up studies. Using the same methodology, the findings from other regions would improve our understanding of cashless society aspirations. Any evidence from different countries, cross-country analyses, or even cross-sectoral analyses within the same country could be an important avenue for further research. Third, this study was focused on examining the change in the use of electronic payments. This is rather a dynamic phenomenon. We have used a cross-sectional analysis to examine the dynamic variable. It is advisable for further studies to provide time-based analyses and capture the evolutionary characteristics of e-payment use.

**Author Contributions:** Conceptualization, M.M.; methodology, M.M., M.O. and M.J.; validation M.O. and M.J.; writing—original draft preparation, M.M., M.O., S.C.K. and M.R.; supervision, S.C.K. and M.J. All authors have read and agreed to the published version of the manuscript.

**Funding:** This research received no external funding.

**Institutional Review Board Statement:** Not applicable.

**Informed Consent Statement:** Informed consent was obtained from all subjects involved in the study.

**Data Availability Statement:** Not applicable.

**Conflicts of Interest:** The authors declare no conflict of interest.

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
