# Peer review of "COVID-19 and Behavioral Factors of e-Payment Use: Evidence from Serbia"

_sustainability, doi:10.3390/su15043188_

Round 1
Reviewer 1 Report
Firstly, I would like to congratulate the authors of the paper “COVID-19 and behavioral factors of e-payment use: Evidence from Serbia” for their efforts and engagement in writing it. The paper is good and deserve being published after few adjustments. Below I will provide the necessary information for the authors in order to make it till the end of the publishing process.
1. First recommendation is to use the name of authors in statements like the one on line 32 (same on line 124, 168 – please check this all around the article). After the name reference can be added of course.
2. Statements such as the one in line 52-54 needs strong references (here is one: https://www.coe.int/en/web/congress/-/kristoffer-tamsons-hate-speech-and-fake-news-detrimental-effect-on-working-conditions-of-local-and-regional-elected-representatives - and the book below text: COUNTERFAKE - A scientific basis for a policy against fake news and hate speech).
3. Line 58 – I suggest cutting down the following words: “not to offend or insult conspiracy theorists, but”.
4. Line 82 also needs references
5. line 130 – please use firstly the extended name of Compound Annual Growth Rate and add CAGR in brackets.
6. lines 332 and 333… p<.00 should be a mistake, please correct.
7. Same on 346
8. Discussion section looks more like a conclusion and vice-versa… maybe integrating it would sound better. Another idea is shifting the content.
9. In regard with the references. Most of them are in good shape; however, I would like to make a strong suggestion to the authors and that is to find inspiration on the MDPI Sustainability journal, there are plenty of great article that could be used to draw this research in an even brighter way. Also try looking at the SCRD Journal (scrd.eu), also some good articles to be found there on one special issue on COVID.
Author Response
Dear reviewer,
Thank you very much for all the comments and suggestions. Please, find attached the document with the explanation of all the corrections that we have done to our manuscript (your comments were addressed in the section REVIEWER 1).
Best,
Authors

Reviewer 2 Report
This paper shows that the Covid pandemic accelerated the use of e-payments.
There is prior work that shows consistent findings. For example, in this paper:
Allen, Franklin, Xian Gu, & Julapa Jagtiani, 2021. "A Survey of Fintech Research and Policy Discussion" Review of Corporate Finance 1 (3-4), 259-339. http://dx.doi.org/10.1561/114.00000007
These authors state at the start of section 10 for example : “The current COVID-19 pandemic has also expedited financial and nonfinancial firms to better serve consumers in the new landscape — by using technology and transforming business models to a fully digital lifestyle and potentially bringing us toward a potential cashless economy.”
I think it would help your arguments to acknowledge that other well-known authors like Franklin Allen agree with your main hypotheses.
In terms of your empirics, I was wondering a few things, including:
- Can you control for additional variables, such as respondent age, gender, wealth, etc?
- Instead of merely stating the coefficient values in the text, can you interpret the economic significance by stating something like “a 1-standard deviation increase in the right hand side variable causes an x% change in the left hand side variable”, or something similar?
It would be great if you could say something more about the representativeness of the data – how many people were contacted, the response rate, potential sample collection biases, etc.
It might be nice to suggest more for future research directions, and policy implications of your study.
Author Response
Dear reviewer,
Thank you very much for all the comments and suggestions. Please, find attached the document with the explanation of all the corrections that we have done to our manuscript (your comments were addressed in the section REVIEWER 2).
Best,
Authors

Round 2
Reviewer 2 Report
I noticed a few typos - please check - e.g. "Other studies implicitly assume infer that COVI-19 pandemics has altered the payment system [67]" should be . "Other studies implicitly assume infer that COVID-19 pandemics has altered the payment system.[67]"
Please check throughout...
Also, I see that the reference [67] appears to be something different - the prior report quoted "The current COVID-19 pandemic has also expedited financial and nonfinancial firms to better serve consumers in the new landscape — by using technology and transforming business models to a fully digital lifestyle and potentially bringing us toward a potential cashless economy.” So your statement about what the [67] paper says is different from what it does say. I would suggest altering the statement.
I hope these comments are helpful.
Author Response
Dear Reviewer,
We have addressed both of your suggestions. Thank you very much for this helpful comments.
Best,
Authors

Round 3
Reviewer 2 Report
Thanks for replying to my prior comments. I hope you found them helpful